# Anthropometric, biochemical, dietary, morbidity and well-being assessments in women and children in Indonesia, India and Senegal: a UKRI GCRF Action Against Stunting Hub protocol paper

Hilary Davies-Kershaw [1] , Umi Fahmida [2] , Min Kyaw Htet [2] , Bharati Kulkarni,[3] Babacar Faye,[4] Dwi Yanti,[2] Dewi Shinta,[2] Nur L Zahra,[2] Tiffany C Angelin,[2] Radhika Madhari,[3] Raghu Pullakhandam [3] , Ravindranadh Palika,[3] Teena Dasi,[3] Sylvia Fernandez Rao,[3] Santosh Kumar Banjara [3] , Kiruthika Selvaraj,[3] Dharani Pratyusha Palepu [3] , Dinesh Yadev,[5] Saliou Diouf,[4] Philomene Lopez-Sall,[4] Babacar Diallo,[4] Princillia Mouissi,[4] Sally Fall,[4] Ibrahima Diallo,[4] Aicha Djigal,[4] Tabitha D Van Immerzeel,[1] Fassia Tairou,[4] Assana Diop,[4] Rebecca Pradeilles,[6,7] Sara Strout,[1] Benjamin Momo Kadia [8] , Darius Tetsa Tata,[5] Modou Lamin Jobarteh [9] , Stephen Allen,[8] Alan Walker [10] , Joanne P Webster [11] , Paul Haggarty,[10] Claire Heffernan,[5] Elaine Ferguson[1]

For numbered affiliations see end of article.

**Correspondence to**
Dr Hilary Davies-Kershaw; Hilary.Davies@lshtm.ac.uk

## ABSTRACT

**Introduction** Child stunting has a complex aetiology, especially in the first 1000 days of life. Nutrition interventions alone have not produced expected impacts in reducing/preventing child stunting, indicating the importance of understanding the complex interplay between environmental, physiological and psychological factors influencing child nutritional status. This study will investigate maternal and child nutrition, health and well-being status and associated factors through the assessment of: (1) anthropometry, (2) biomarkers of nutrition and health status, (3) dietary intakes, (4) fetal growth and development, (5) infant morbidity, (6) infant and young child feeding (IYCF) and (7) perinatal maternal stress, depression and social support.

**Methods** This study will be conducted in a prospective pregnancy cohort in India, Indonesia and Senegal. Pregnant women will be recruited in the second (Indonesia, Senegal) and third (India) trimester of pregnancy, and the mother and infant dyads followed until the infant is 24 months of age. During pregnancy, anthropometric measures will be taken, venous blood samples will be collected for biochemical assessment of nutrition and health status, dietary intakes will be assessed using a 4-pass-24-hour dietary recall method (MP24HR), fetal ultrasound for assessment of fetal growth. After birth, anthropometry measurements will be taken, venous blood samples will be collected, MP24HR will be conducted, infant morbidity and IYCF practices will be assessed and a sample of breastmilk will be collected for nutrient composition analyses. Perinatal maternal stress, depression, social

## WHAT IS ALREADY KNOWN ON THIS TOPIC

⇒ The global burden of child stunting remains unacceptably high.
⇒ While some progress has been made, reduction of the burden of child stunting is very slow and sometimes unresponsive.
⇒ Increasing the understanding of the complex aetiology of child stunting is critical to programmes and policies to further prevent or reduce the global burden.

## WHAT THIS STUDY ADDS

⇒ Discern the contribution of maternal and child nutrition and nutritional status on child stunting as part of an interdisciplinary analysis.
⇒ Increase understanding of complex inter-relationships and biological factors that influence maternal and child nutrition and stunting.
⇒ Contribute to an understanding of the influence of maternal depression, maternal stress and social support on child nutritional status.

## HOW THIS STUDY MIGHT AFFECT RESEARCH, PRACTICE OR POLICY

⇒ Inform policy-making by determining and comparing risks of undernutrition across the three settings using anthropometric, biochemical and dietary nutritional status for pregnant women and their offspring.

support and hair cortisol levels (stress) will be measured. The results from this study will be integrated in an interdisciplinary analysis to examine factors influencing infant growth and inform global efforts in reducing child stunting.

**Ethics and dissemination** Ethical approval was granted by the Ethics Committee of the London School of Hygiene and Tropical Medicine (17915/RR/17513); National Institute of Nutrition (ICMR)-Ministry of Health and Family Welfare, Government of India (CR/04/I/2021); Health Research Ethics Committee, University of Indonesia and Cipto Mangunkusumo Hospital (KET-887/UN2.F1/ETIK/PPM.00.02/2019); and the Comité National d'Ethique pour la Recherche en Santé, Senegal (Protocole SEN19/78); the Royal Veterinary College (URN SR2020-0197) and the International Livestock Research Institute Institutional Research Ethics Committee (ILRI-IREC2020-33). Results will be published in peer-reviewed journals and disseminated to policy-makers and participating communities.

## INTRODUCTION

Adequate nutrition during the first 1000 days of life (from conception to 24 months of age) is essential for optimal linear and ponderal growth, immune function and cognitive development.[1 2] Poor maternal nutritional status during pregnancy and lactation contributes to infant undernutrition through its influence on infant birth weight and length, gestation length, infant immunity and the nutrient content of breastmilk.[1 3] Important intermediate factors contributing to undernutrition after birth include the quality of infant care and breastfeeding practices, infant's dietary intake, and the frequency and severity of infant illnesses.[4] Along with the widely recognised underlying causes of undernutrition, including food security, poverty, maternal education and health service infrastructure,[4] there is a growing interest in less-researched areas such as maternal depression, stress and social support, which can influence infant care and feeding practices.[4 5] The Action Against Stunting Hub observational Study (AASH) will follow a cohort of mother–infant dyads from pregnancy to 24 months post partum in Hyderabad, India, East Lombok, Indonesia and Kaffrine, Senegal where biological and environmental exposures and the prevalence of child-stunting differ, ranging from 26% in Kaffrine District (Senegal), 28% in Telegana State, (India) to 44% in East Lombok, (Indonesia).[6–8]

This study will investigate the contribution of factors affecting child growth and development through assessment of:

1. Anthropometric status of pregnant women, their infants and the infants' biological father.
2. Biochemical and dietary nutritional status of women during pregnancy and infants from 6 months of age.
3. Fetal growth and development.
4. Infant and young child feeding (IYCF) practices and morbidity of the infants.
5. Perinatal maternal stress, depression and social support.
6. Associations of environmental, biological and sociological factors on infant growth.

## METHODS AND ANALYSIS

Prospective pregnancy cohort studies will be conducted in Hyderabad, India (n=717), East Lombok, Indonesia (n=702) and Kaffrine, Senegal (n=763). The sample size and power were calculated to test 100 independent epigenetic signatures between stunted and non-stunted children (Epigenetic Studies in Children at Risk of Stunting and their Parents in India, Indonesia and Senegal: a UKRI GCRF AASH protocol paper BMJ Paeds Open Ramsteijn *et al*, in press). Women will be recruited in the second (Senegal and Indonesia) or third (India-delays due to COVID-19) trimester of pregnancy. Their infants will be followed until 24 months of age. The selection of methods for the nutrition study was guided by a conceptual framework for stunted linear infant growth (figure 1). Summaries of timings of data collection and measurement indicators and timings of data collection are presented in online supplemental tables 1 and 2, respectively. All data, except dietary intake in India and Senegal, will be recorded using a centralised data collection application (CommCare, Dimagi, Cambridge, Massachusetts, USA).

### Data collection
#### Anthropometric assessment

Anthropometric measurements of growth and body composition will be conducted on pregnant women in the second and/or third trimester of pregnancy, from the infants' biological father, and on infants within 72 hours after birth and at 3, 6, 9, 12, 18 and 24 months of age by trained enumerators (online supplemental table 2). Measurements will be done in duplicate, using standardised procedures and calibrated equipment, with a third measurement taken if the first and second measurements do not agree with the predefined limits (table 1).[9]

Weight will be measured using the SECA-876 scale with a mother–child function (range=0–250 kg, precision±0.1 kg). Adult height (standing and sitting) and infant length will be measured using a portable UNICEF-height/length board (range=0–221 cm, precision±0.1 cm), and for sitting height using a purpose-built anthropometric box.[10] Mid-upper arm circumference and infant head circumference will be measured using the Lufkin-WP601 tape-measure. Both infant and adult arm measurements will be done on the left-hand side. Knee-to-heel length will be measured using a segmometer (Malby labs-India and Indonesia and Sealy-Senegal). Skinfolds measurements will be taken using a Holtain skinfold-calliper (range 0–40 mm, precision±0.2 mm).

In the data cleaning process consistency checks in CommCare for infant z scores; length-for-age (<−6 or >6 SD), weight-for-length (<−5 or >5 SD) and weight-for-length (<−6 or >5 SD),[9 11] and for a negative difference between the current and preceding measurement of infant weight and length will be done. Z-scores will be calculated using the WHO child growth standards software.[12] Prior to data collection, in each country, a standardisation session will take place to calculate

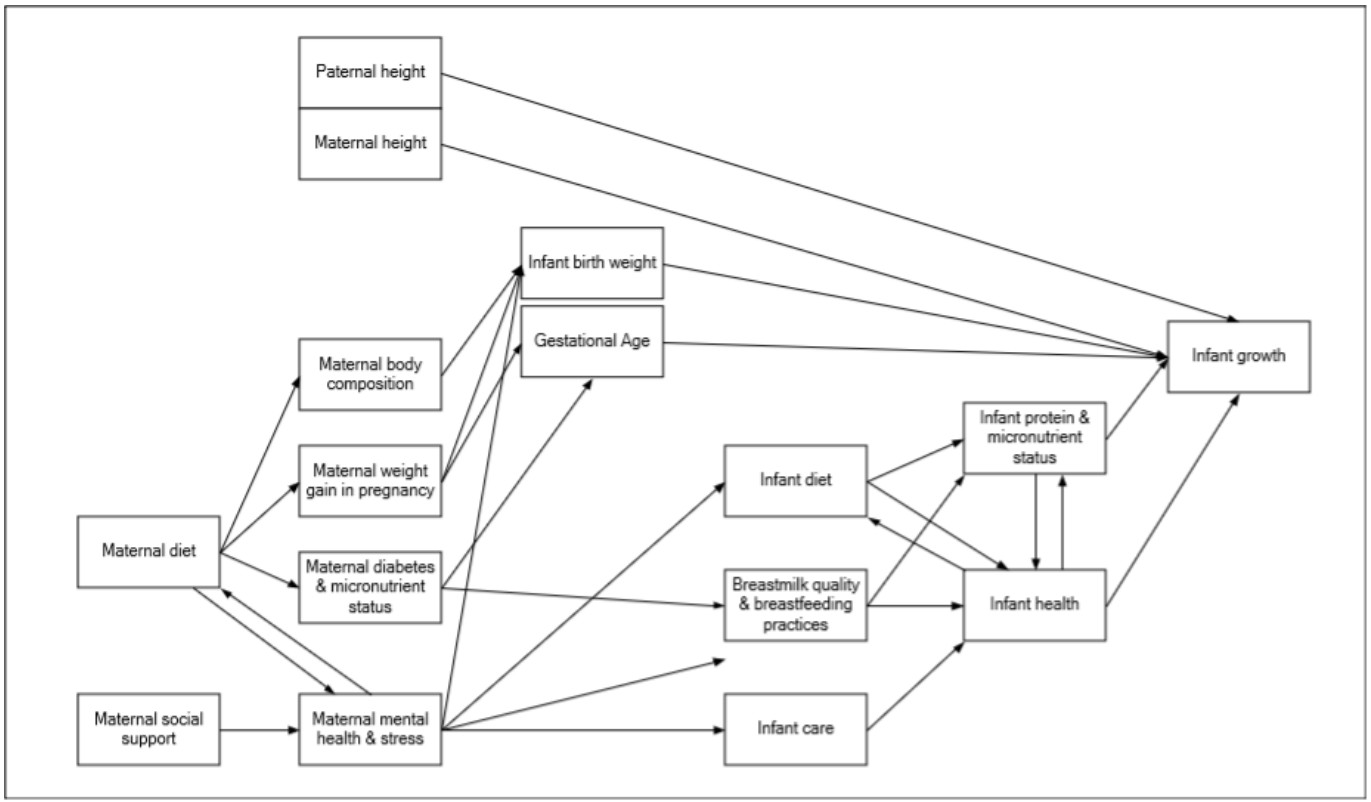

**Figure 1** A conceptual framework of pathways to assess the impact of nutrition, nutritional status and maternal wellbeing on child stunting.

intraobserver and interobserver absolute and relative technical error of measurements and coefficients of reliability for all anthropometric measurements.[9 11] Regular refresher training sessions will be conducted and, field supervisors will monitor data collection, using a checklist based on the study's Standard Operating Procedures (SOP).

### Biochemical assessment of blood samples

Using country-specific standard procedures,[13] blood samples will be collected from pregnant women (ie, 10 mL in India and Senegal and 7 mL in Indonesia) during the third trimester (and second trimester in a subsample of women enrolled in a concurrent egg intervention trial

| Table 1 Predefined limits in the difference between first and second anthropometry measurements | |
|---|---|
| **Measure** | **Predefined limits** |
| Adult and infant weight | 0.1 kg |
| Adult height | 0.5 cm |
| Adult mid upper arm circumference | 0.5 cm |
| Adult skinfolds | 0.2 cm/2 mm |
| Infant length | 0.7 cm |
| Infant head circumference | 0.5 cm |
| Infant knee to heel | 0.5 cm |
| Infant mid upper arm circumference | 0.5 cm |
| Infant skinfolds | 0.2 cm/2 mm |

in Indonesia). In Infants, 3 mL in India and Senegal and 2.5 mL in Indonesia of non-fasting blood samples will be collected at 6 months of age (and at 18 months in a subsample of children enrolled in an egg intervention study in India). The samples will be collected using Becton-Dickinson Vacutainer (EDTA and trace-element free). A complete blood count and glycated haemaglobin (women only) will be analysed on whole blood samples within 4–8 hours of collection. Blood samples will then be processed for separation of plasma, red blood cells and serum and aliquoted into microtubes for storage. All aliquots will be stored at −80°C until analyses (table 2). The time of blood sample collection and separation will be recorded. The risks of contamination/analyte deterioration will be minimised by wearing powder-free gloves, keeping samples cool and protection from light.

Blood samples will be analysed for maternal and infant biochemical indicators of nutritional status, inflammation and growth factors in infants (online supplemental table 2). Certified reference materials (CRMs) will be included in the analyses where available to ensure analytical accuracy, precision and reliability. A certified pooled plasma sample (Human Plasma Pool (H100500) from European Pharmacopoeia for Indonesia) will also be analysed across all laboratories to ensure intercountry comparability and analytical reliability.

### Dietary intakes

A 4-pass 24-hour dietary recall (MP24HR)[14] will be used to assess dietary intake. During the second (Indonesia

**Table 2** Analysis methods for parameters in India, Indonesia and Senegal

| Parameters | India | Indonesia | Senegal |
|---|---|---|---|
| Blood samples | | | |
| Complete blood count (CBC) Glycated haemaglobin (HbA1c-pregnant women only) | CBC auto-analyser (Horiba ABX Micros ES60) HbA1c analyser (Affinion Hb1Ac, Abbot) | CBC auto-analyser (SYSMEX XP100) HbA1c analyser (PocketChem A1c, Arkray) | CBC auto-analyser HbA1c analyser (HPLC D10 Biorad) |
| Red blood cell folate Methylmalonic acid Choline Betaine | LC-MS/MS | LC-MS/MS | LC-MS/MS |
| Homocysteine Vitamin D Vitamin $B_{12}$ Vitamin $B_9$ | LC-MS/MS | LC-MS/MS | CLIA |
| Red blood cell-fatty acids | GC-FID | GC-FID | GC-FID |
| Amino acids | HPLC | LC-MS/MS | LC-MS/MS |
| Retinol | HPLC | HPLC | HPLC |
| Ferritin TfR | ELISA | ELISA-Quansys multiplex system | CLIA ELISA |
| C reactive protein $\alpha$1-Acid glycoprotein Hepcidin Retinol binding protein Insulin like growth factor-1 Insulin-like growth factor binding protein Intestinal fatty-acid binding protein | ELISA | ELISA-Quansys multiplex system | ELISA-Quansys multiplex system |
| Minerals/trace elements | ICP-MS | ICP-MS | ICP-MS |
| Remaining Red blood cells Remaining plasma | Kept at −80°C | Kept at −80°C | Kept at −80°C |
| Breastmilk samples | | | |
| Fat-soluble vitamins A and D | HPLC/LC-MS | HPLC/LC-MS | HPLC/LC-MS |
| Water-soluble vitamins $B_1$, $B_2$, $B_3$, $B_6$, $B_9$ and $B_{12}$ | LC-MS | LC-MS/MS | LC-MS |
| Fatty acids | GC-FID | GC-FID | GC-FID |
| Minerals | ICP-MS | ICP-MS | ICP-MS |
| Oligosaccharides | SPEC and HPLC | LC-MS/MS | SPEC and HPLC |
| Remaining breastmilk | Kept at −80°C | Kept at −80°C | Kept at −80°C |
| Hair samples | | | |
| Cortisol | ELISA | LC-MS/MS | ELISA |

CLIA, chemiluminescence immunoassay; GC-FID, gas chromatography-flame-ionisation detection; HPLC, high performance liquid chromatography; ICP-MS, inductively coupled plasma mass spectrometry; LC-MS/MS, liquid chromatography with tandem Mass spectrometry; SPEC, solid phase extraction chromatography.

and Senegal) and third trimesters (all sites), women will be asked to recall all foods and beverages consumed in a preceding 24 hours. The infants' dietary intakes from complementary foods will be estimated using an interactive 24-hour recall at 6, 9, 12, 18 and 24 months of age. For both mothers and children, a repeat recall will be collected on a randomly selected 10% subsample of the population to estimate intrasubject variability in nutrient intakes. The nutrient density of the complementary foods will be compared with a recommended nutrient density.[15] For children ≥12 months of age, breast milk intakes will be estimated as the difference between mean estimated energy requirements and the mean energy intakes from complementary foods to estimate total daily intakes of energy and nutrients. The percentage of mothers and 1-year-old children at risk of inadequate nutrient intakes

will be estimated, depending on the nutrient, using the Estimated Average Requirement fixed cut-point approach or the full probability approach Quantities of foods and drinks consumed will be estimated by weighing actual foods/beverages or food models using calibrated digital electronic kitchen scales (Salter, 1024,-India and Senegal and Tanita, KD-160-Indonesia; precision±1.0g), dimension measurements (cm), size estimations (small, medium or large) or using photographs, depending on the item. Details on sources of food acquisition will be recorded, and questions will be asked about whether their food intake on the preceding 24 hours was typical of their usual intakes, and for infants, whether they were breastfed.

Local conversion factors to convert portion size estimates to grams consumed and average recipe data to convert the quantity of a mixed-dish consumed into grams of individual ingredients will be developed when they are not available from previous studies.

In Indonesia, MP24HR data will be entered on a tablet (CommCare-application),[16] while in India and Senegal, a paper form will be used to collect dietary data from respondents and entered on a web-based application created specifically for this project and then uploaded on to the study data repository. Both applications (paper and tablet format) will calculate each participant's intakes of energy and nutrients using the local conversion factors, average recipe data and country-specific food composition tables (FCT) in India and Indonesia,[16 17] and an FCT developed using a hierarchy of other food composition values in Senegal.[18–24] Paper-based forms will be reviewed immediately after data collection by the field supervisors and any questions resolved.

## Ultrasound

Pregnant women in all three countries will have an antenatal ultrasound in government-recognised diagnostic centres at 32 weeks (±2 weeks) to evaluate fetal growth and development.[25] A trained obstetric sonographer will use a ultrasound system,[26] to take the following measurements in triplicate, which will be checked by a radiologist; biparietal diameter, occipito-frontal diameter, head circumference, anteroposterior abdominal diameter, transverse abdominal diameter, abdominal circumference and femur length, fetal presentation, placental localisation and amniotic fluid volume, fetal movement, cardiac pulsation and the presence of congenital anomalies.[27] Estimated gestational age will be calculated using the crown-rump length.[28] The ultrasound report will be shared with the participant and advised to see the on-call doctor if necessary.

## IYCF practices during the first 6 months after birth and breastmilk nutrient composition

Within 72 hours of birth, enumerators will administer a questionnaire to mothers to evaluate: (1) prelacteal feeding (other feeds, besides breastmilk), (2) breastfeeding initiation and timing, (3) colostrum fed to the infant, (4) other feeds, besides breastmilk and (5) infant nutrition supplements.[29 30] Once a month until the infant is 6 months of age, the enumerator will ask mothers whether their infant has consumed anything other than breastmilk during the previous 4 weeks. If the infant was not exclusively breastfed, detailed questions about the type of newly introduced liquids or solids will be asked.

A morning (between 6:00 and 12:00 hours) breastmilk sample will be collected from mothers at 3 months post partum by asking them to express all milk from one breast into a sterile collection bottle, using a manual breastmilk pump, following standard procedures.[31] After collection, the bottles will be placed in an ice-box and transported to a laboratory, where milk samples will be mixed with a vortex-mixer and aliquot 1 mL in amber 1.5 mL tube (fatty acid, fat and water-soluble vitamin analysis-Indonesia only), 10 mL into three amber-coloured screw cap 15 mL Eppendorf-tubes (fat-soluble and water-soluble vitamins and fatty acids analyses for India and Senegal), one clear trace-element free tube (minerals analysis for India and Senegal) and 1 mL in clear 1.5 mL Eppendorf-tubes (oligosaccharides analysis for all three countries).[30] Samples will be immediately stored at −80°C (table 2). CRM (NIST-1549a)—a standard control sample, will be included in the analyses to ensure analytical accuracy, precision and reliability.

## Child morbidity

Child morbidity will be recorded using an adapted country-specific pictorial diary from birth to 24 months of age.[32] Mothers will be asked to complete the diary daily in India and Indonesia and collected weekly. Enumerators will help complete any incomplete forms. In Senegal, morbidity data will be collected weekly via enumerator-led telephone interviews. The same information will be collected across all three countries. Mothers will be asked if her child was healthy or had active, reduced appetite, diarrhoea (with or without blood and/or mucus in the stools), vomiting, bloated stomach, fever, cough (with or without nasal discharge and/or wheezing), ear pain, chicken pox or measles.[32] They will also record any health centre visits and any medication the infant received. Additional morbidity questions (1-week recall) will also be asked at the same time as monthly IYCF questionnaire up to 6 months of age.

## Maternal mental health and social support

Maternal stress will be assessed using cortisol level in hair samples,[33] and the Global Measure of Perceived Stress Scale (GMPSS) questionnaire.[34] Hair samples will be collected and GMPSS questionnaire administered during pregnancy (ie, second trimester in Indonesia and Senegal and third trimester in India) and at 3 and 9 months post partum. Hair grows approximately 1 cm/month, therefore, 3 cm of hair will be collected as close as possible to the scalp from the posterior vertex of the head, placed on a hair collection card indicating the root end, wrapped in aluminium foil and stored in a dry,

dark environment at room temperature until analysed, as described elsewhere.[33 35] Information on hair-care practices, which influence the hair-cortisol measurements, will also be recorded.[35] The GMPSS consists of 14 questions, on thoughts and feelings, which participants will rate using a 5-point Likert scale.[34]

Maternal depression will be measured using the Edinburgh Postnatal Depression Scale (EPDS) and the Self-Reporting Questionnaire (SRQ) during the second (Senegal and Indonesia) and third trimester (India), and then in all countries at 3, 9 and 18 months post partum.[36 37] The EPDS consists of 10 statements, where participants are asked to rate how they felt in the last 7 days on a 4-point Likert scale.[36] As well as the classic EPDS, the new shorter version of the EPDS will also be asked to validate it against the classic EPDS in India and Senegal.[38] The SRQ was developed by the WHO to screen for common mental disorders.[39] It consists of 20 questions with dichotomous yes/no answers.

Maternal social support will be assessed using the Multidimensional Scale of Perceived Social Support,[40] at 3, 9 and 18 months after birth. It includes 12 questions where participants are asked to rate how strongly they agree or disagree on a 7-point Likert scale,[40] and their responses are recorded.

### Data collected by other workstreams within the AASH study

The AASH incorporates interdisciplinary workstreams and will be collecting data on the following; detailed household questionnaires, epigenetics (mother, father and infant saliva samples), gut health and the microbiome (mother and infant stool samples), Water, Sanitation and Hygiene in the home environment (household observations), food systems (market and household observations and surveys) and education and cognition (household and preschool observations).

### Statistical analysis plan

Data will first be analysed at country level and second pooled together where appropriate. The primary outcomes are: (1) growth velocity between time points from birth to 18 months and (2) height for age z scores at time points 3, 6, 9, 12 and 18 months. Based on the literature, we have decided a priori to include the following parental covariates in our models; age, cast (India only), village/health post, socioeconomic status, marital status, current work status and education status. Multivariable statistical models will be developed using directed acyclic graphs and related approaches including structural equation modelling with preanalysis to understand confounders and mediators on the pathway between exposures and outcomes. We will build our models and adjust for confounders using a forward stepwise approach with Akaike information criterion to identify the model of best fit.[41] All analysis will be conducted by using STATA V.17 and R.

### Patient and public partnership statement

Patients and public were not involved in the design of this study.

## DISCUSSION

Addressing early life undernutrition is a public health priority in LMICs given its detrimental impacts on child survival, growth and development. The full extent and aetiology of undernutrition, however, are not well established, given the limited number of cohort studies from fetal life through to early childhood. As such studies are needed to understand the complex interplay among environmental and intrinsic physiological factors influencing early life nutritional status in different contexts.

This study and others within the AASH will investigate these complex relationships across three distinct geographies in sub-Saharan Africa and Asia. The analysis will examine factors affecting infant growth through three distinct periods; fetal period, 0–6 months of age and during the complementary feeding (6–24 months of age). The selection of its data collection protocols is underpinned by a theoretical framework of factors influencing infant and childhood growth (figure 1), which will be further expanded through AASH interdisciplinary collaborations to include influences of epigenetics, gut health and food systems. This study will also contribute to less-researched areas such as the influence of maternal depression, maternal stress and social support on infant care and feeding practices which ultimately affect infant growth.

There are several limitations to this study. First, it was not feasible for this cohort study to include the preconception period given the time frame and resources available. Also, recruitment in the first trimester would not have been appropriate given cultural sensitivities, in some participating countries, of admitting pregnancy in the first trimester. The collection of data during the second and third trimester of pregnancy, however, will provide a wealth of relevant data related to exposures during pregnancy that will influence fetal and later infant growth, including a measure of weight gain in pregnancy, stress (hair cortisol reflects stress for the past 3 months) and maternal dietary intakes. Second, although identical indicators of micronutrient status will be analysed across countries, the methods of analyses for two biochemical indicators differ. To minimise the influence of interlaboratory assay differences on cross-country comparisons, a pooled sample and CRMs, where available, will be analysed in all laboratories. Third, in our cohort, since we enrolled the participants in the second (Indonesia and Senegal) and the third trimester (India) of pregnancy, the study teams could only carry out ultrasound scans during these time points rather than the first trimester. However, data from the first and second trimester scans are available in around 80% and 90% participants in India, respectively, and will be used for gestational age estimation if available. If not, the date of last menstruation will be used. Studies have shown that routine ultrasound scans at around 32 weeks will help in detection of early intrauterine growth restriction (IUGR).[42] Therefore, within our limitations to perform the ultrasound scan at second and third trimesters as a part of the study

protocol, we think these ultrasound measurements will be useful to identify early IUGR by comparison with INTERGROWTH-21 charts. This information, coupled with the infant growth data, will be valuable because such data are scarce in a low-resource setting.[43]

Fourth, interpopulation differences in maternal literacy necessitated a change in morbidity data collection protocols in Senegal. Lastly, when investigating causal inferences, there are limitations with a prospective cohort study design, including lost to follow-up bias and confounding. These biases will be minimised through concerted sensitisation/motivational efforts to reduce lost to follow-up and the use of DAGS.

## Conclusions

Results from the nutrition study conducted in three diverse geographic regions will form part of the 'whole child approach' pursued under AASH. The interdisciplinary collaborations established across the hub provide a unique opportunity to investigate the complex interrelationships among environmental and intrinsic physiological factors influencing early life growth and development. Such insights are important for informing effective intervention design and the policy actions needed to support global efforts in reducing early child stunting.

**Author affiliations**

[1]Department of Population Health, London School of Hygiene & Tropical Medicine, London, UK

[2]Regional Centre for Food and Nutrition, SEAMEO, University of Indonesia, Jakarta, Indonesia

[3]Deparments of Maternal and Child Health and Dietetics Division, National Institute of Nutrition, Hyderabad, India

[4]Service de Parasitologie-Mycologie- Pédiatrie, Faculté de médecine, UCAD, Dakar, Senegal

[5]Department of Public Health, London International Development Centre, London School of Hygiene & Tropical Medicine, London, UK

[6]School of Sport, Exercise and Health Sciences, Loughborough University, Loughborough, UK

[7]UMR, MOISA, Montpellier, France

[8]Department of Clinical Sciences, Liverpool School of Tropical Medicine, Liverpool, UK

[9]Department of Epidemiology & Population Health, London School of Hygiene & Tropical Medicine, London, UK

[10]The Rowett Institute, University of Aberdeen, Aberdeen, UK

[11]Department of Pathobiology and Population Sciences, University of London, London, UK

**Contributors** HD-K and EF were responsible for the overall design, training and overseeing implementation of the research. UF, MKH, BK, BF, RM, RPullakhandam, RPalika, TD, SFR, SD, RPradeilles, SA, AW, JPW, PH and CH were involved in its design. UF, MKH, BK, BF, DY, DS, NLZ, TCA, RM, RPullakhandam, RPalika, TD, SFR, SKB KS, DPP, DY, SD, PL-S, BD, PM, SF, ID, AD, TDVI, FT, AD, SS, BMK and DTT implemented the research. HD-K and EF wrote the manuscript. All authors read, provided comments on and approved the final version of the manuscript.

**Funding** This study is funded by UKRI Global Challenges Research Fund (GCRF) (No. MR/S01313X/1).

**Disclaimer** The views expressed in this work are those of the authors and not the funder.

**Competing interests** none.

**Patient and public involvement** Patients and/or the public were not involved in the design, or conduct, or reporting, or dissemination plans of this research.

**Patient consent for publication** Not applicable.

**Ethics approval** This study involves human participants and was approved by the Ethics Committee of the London School of Hygiene and Tropical Medicine (17915/RR/17513); National Institute of Nutrition (ICMR)- Ministry of Health and Family Welfare, Government of India (CR/04/I/2021); Health Research Ethics Committee, University of Indonesia and Cipto Mangunkusumo Hospital (KET-887/UN2.F1/ETIK/PPM.00.02/2019); and the Comité National d'Ethique pour la Recherche en Santé, Senegal (Protocole SEN19/78); the Royal Veterinary College (URN SR2020-0197); and the International Livestock Research Institute Institutional Research Ethics Committee (ILRI-IREC2020-33).

**Provenance and peer review** Not commissioned; externally peer reviewed.

**Data availability statement** Data sharing is not applicable to this article as no new data were created or analyzed in this study.

**ORCID iDs**

Hilary Davies-Kershaw http://orcid.org/0000-0002-2044-2469
Umi Fahmida http://orcid.org/0000-0003-1403-6242
Min Kyaw Htet http://orcid.org/0000-0001-6417-2942
Raghu Pullakhandam http://orcid.org/0000-0002-3758-667X
Santosh Kumar Banjara http://orcid.org/0000-0002-0893-9552
Dharani Pratyusha Palepu http://orcid.org/0000-0003-1692-020X
Benjamin Momo Kadia http://orcid.org/0000-0002-8566-7132
Modou Lamin Jobarteh http://orcid.org/0000-0002-7350-6980
Alan Walker http://orcid.org/0000-0001-5099-8495
Joanne P Webster http://orcid.org/0000-0001-8616-4919

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
