## [Reviewer comments · BMJ Paediatrics Open]

ARTICLE DETAILS

TITLE (PROVISIONAL)	Anthropometric, biochemical, dietary, morbidity and wellbeing assessments in women and children in Indonesia, India and Senegal: a UKRI GCRF Action Against Stunting Hub protocol paper
AUTHORS	Davies-Kershaw, Hilary Fahmida, Umi Htet, Min Kyaw Kulkarni, Bharati Faye, Babacar Yanti, Dwi Shinta, Dewi Zahra, Nur Angelin, Tiffany C. Madhari, Radhika Pullakhandam, Raghu Palika, Ravindranadh Fernandez Rao, Sylvia Selvaraj, Kiruthika Palepu, Dharani B, Santosh Kumar Yadev, Dinesh Diouf, Saliou Lopez-Sall, Philomene Diallo, Babacar Mouissi, Princillia Fall, Sally Diallo, Ibrahima Djigal, Aicha Immerzeel, Tabitha Tairou, Fassia Diop, Assana Pradeilles, Rebecca Strout, Sara Momo Kadia, Benjamin Tata, Darius Jobarteh, Modou Allen, Stephen Walker, Alan Webster, Joanne P. Haggarty, Paul Heffernan, Claire Ferguson, Elaine

VERSION 1 - REVIEW

REVIEWER	Reviewer Name: Dr. Peter Flom Institution and Country: Peter Flom Consulting, United States
REVIEW RETURNED	26-Sep-2022

GENERAL COMMENTS	I mostly confine my remarks to statistical aspects of this paper. Unfortunately, I see some fairly major flaws. Fortunately, though, this is a protocol paper, and the flaws can be corrected. My first question is more methodological than statistical. Why these three sites? And why start at different trimesters in different sites? p. 5 line 28-30 Why will these be done only in India and Senegal? What will be done if the consistency check is not met? Why will some measures be done at 6 times in the child's life and others at only 2? And some maternal measures at 3 time points? Why not be consistent? p. 6 lines 25-27 How will dietary intake of infants be measured? Surely most of these babies will be breast feeding. How will you figure out how much milk they get? There are other differences in method between countries that need to be explained. The same methods should be used everywhere, unless there is good reason not to do so. Using different methods adds noise. (Table 1 shows the differences). p 9 Why not determine the confounders and effect modifiers now? (And what's the difference between a confound and an effect modifier? AFAIK, a confounder IS an effect modifier). What exactly is/are the dependent variables? It looks like "growth" is one and "morbidity" possibly several more. These would need different methods. But Table 2 indicates a whole lot of measures, without clarifying which are dependent and which are covariates. Also, there could easily be nonlinear effects and interactions. Also, there needs to be a power analysis and discussion of sample size.
--

REVIEWER	Reviewer Name: Dr. Peter Rohloff Institution and Country: Guatemala
REVIEW RETURNED	18-Oct-2022

GENERAL COMMENTS	This cohort study will attempt to use multiple data streams to look at factors associated with child stunting in three countries. The strength of the study is an integrated protocol in three countries. I think there are some unacknowledged weaknesses and methodological issues that could be clarified however. 1. There are no details on the size of the pregnancy cohort, and no power calculations. It is unknown what are the primary outcomes of interest and if there is sufficient sample size to observe meaningful differences in the outcomes of interest (either pooled across all three sites or at the country level). Two of the country sites have relatively low rates of stunting, which also raises concern about the adequacy of the sample and number of observations to see what needs to be seen. 2. The statistical analysis plan is vague - for example will the team be building models with a dependent variable of stunted/not
---

	stunted? or continuous anthropometric indicator? The authors also say they will develop DAGs as part of their plan, but these should already be developed and presented, else there is a risk that some important data element will be left out during data collection! 3. vlf authors plan to conduct repeat measures assessments of anthropometric data, then attention needs to be given to the difficulty with especially height-for-age Z scores analyzed within subjects over time. Height for age difference is better. Jef Leroy and others have written extensively about this and it should be addressed. 4. Although the authors are planning a lot of diverse data, most of it remains within fairly traditional nutritional/anthropometric categories. If the goal is to contribute definitely to the conversation of stunting then the planned data streams are unlikely to significantly move the needle. What about microbiome? Intestinal mucosal dysfunction/ EED? Environmental toxins like aflatoxin? These are all active areas of investigation in the stunting field right now but are not included here. There also doesn't appear to be any plan to address WASH determinants, a major weakness. 5. The major determinant of stunting is intergenerational/epigenetic effects. The plan to investigate this here is very limited, as far as I can tell only parental heights will be recorded. No DNA methylation or similar studies are proposed - this omission should be addressed. 6. From the perspective of intergenerational transmission of stunting, the timing for recruitment of the pregnancy cohort is also much too late. Opportunities to investigate preconceptional maternal determinants are missed, as is access to nutrients, adequate weight gain, and medical care during the critical early pregnancy period. 7. I have a major concern with the ultrasound protocol. I assume the primary goal here is to assess for fetal growth restriction. However, since the cohort is being recruited late in the second or third trimester, and no details are provided about pregnancy prior to that period, the value of this assessment is very limited. It is quite likely that many women will not know the date of the last menstrual period accurately. Without a first trimester dating ultrasound, and without accurate dates and scrupulous first trimester pregnancy care, the value of a later fetal ultrasound is very limited and will result in significant under- and over-estimation of fetal age/size. The mention of crown-rump length would also be irrelevant in a third trimester ultrasound. 8. For dietary assessments, it is unclear if efforts will be made to tabulate infant diet indicators in a way that would permit comparison to published WHO IYCF indicators.
--	---

VERSION 1 – AUTHOR RESPONSE

Dear Editor of BMJ Pediatrics

Thank for you considering our manuscript and for sending it out for peer-review. Please find our response to comments from the reviewers below.

Dear Reviewer 1

Thank you for peer-reviewing our protocol paper. Although this is a protocol paper, some of the work has already started due to strict timing of funding, so we will aim to respond accordingly.

1) My first question is more methodological than statistical. Why these three sites? And why start at different trimesters in different sites?

Thank you for this comment. This protocol paper is part of a collection of protocol papers that make up the Action Against Stunting Cohort Study. All these protocol papers, which are made up of different workstreams, are currently being submitted to the BMJ Pediatrics journal supplement. The rationale for choice of sites and the different start time-points will be included in the main protocol paper (Heffernan, et al).

We have included the reference to this paper in line 13 on page 4.

2) p. 5 line 28-30 Why will these be done only in India and Senegal? What will be done if the consistency check is not met?

Thank you for this comment, consistency checks were planned and so were in the original protocol paper but have proved logistically challenging to implement so will now be done in all three countries as part of the data cleaning process. The text has been modified in line 28 on page 5 as follows: "In the data cleaning process consistency checks in CommCare for infant z scores; length-for-age (<-6 or >6sd), weight-for-length (<-5 or >5sd) and weight-for-length (<-6 or >5sd)(10,12), and for a negative difference between the current and preceding measurement of infant weight and length will be done."

3) Why will some measures be done at 6 times in the child's life and others at only 2? And some maternal measures at 3 time points? Why not be consistent?

Thank you for this comment. For the infant it was decided to take the full set of anthropometry measures every three months to reduce participant burden and cost. Most participants will need to travel to the health centre. The enumerators will be visiting the mothers and their infants each month between 1-6 months to collect information on breastfeeding and morbidity. This was seen as an opportunity to also collect monthly infant and mother weight. It was decided to only collect one blood sample for both mother and infant to reduce the stress of taking blood, especially for the infant and the cost of analysing the blood.

4) p. 6 lines 25-27 How will dietary intake of infants be measured? Surely most of these babies will be breast feeding. How will you figure out how much milk they get?

For clarity the sentence has been edited in line 26, page 6 as follows: "The infants' dietary intakes from complementary foods will be estimated using an interactive 24-hour recall at 6, 9, 12, 18 and 24 months of age. For both mothers and children, a repeat 24-hour recall will be collected on a randomly selected subsample of the population (i.e., approximately 10%) to estimate intra-subject variability in nutrient intakes."

We have also added the following sentence to show how we will be estimating the amount of breastmilk the infants would be receiving.

Line 32, page 6: "The nutrient density of the complementary foods will be compared to a desired nutrient density (Dewey & Brown, 2003). For children ≥ 12 mo. of age, breast milk intakes will be estimated as the difference between mean estimated energy requirements and the mean energy intakes from complementary foods to estimate total daily intakes of energy and nutrients. The percentage of mothers and 1-year old children at risk of inadequate nutrient intakes will be estimated, depending on the nutrient, using the EAR fixed cut-point approach or the full probability approach. "

5) There are other differences in method between countries that need to be explained. The same methods should be used everywhere, unless there is good reason not to do so. Using different methods adds noise. (Table 1 shows the differences).

In table 1, the methods are the same, however, there are differences in some of the timings due to the later start date in the India site These are due to the study in India having to start in the third trimester instead of the second trimester. These will be taken into consideration during analysis.

6) p 9 Why not determine the confounders and effect modifiers now? (And what's the difference between a confound and an effect modifier? AFAIK, a confounder IS an effect modifier).

The main protocol paper (Heffernan, et al) will include a detailed section on statistical analysis. We have therefore decided to remove the statistical analysis section in this paper and will refer to the main protocol paper.

We have added the following to line 19, page 9

The statistical analysis of the Action Against Stunting Hub Study is explained in detail in the main protocol paper (Heffernan, et al)

7) What exactly is/are the dependent variables? It looks like "growth" is one and "morbidity" possibly several more. These would need different methods. But Table 2 indicates a whole lot of measures, without clarifying which are dependent and which are covariates.

Thank you for this comment. The primary outcomes are length, length -for-age z score and velocity-for-length z score. These have been indicated in Table 2 We have also included if the variables will be continuous, binary or categorical. There will be more information in the paper by Jobarteh, et al on the independent and dependent variables.

8) Also, there could easily be nonlinear effects and interactions.

Thank you for this comment. This is true, but these non-linear effects and interactions will only be identified when analysing the data known. The fact that these will be identified will be included in the main protocol paper by Jobarteh et al.

9) Also, there needs to be a power analysis and discussion of sample size.

This will be included in the main protocol paper (Heffernan, et al)

Dear Reviewer 2

Thank you for peer-reviewing our protocol paper. Although this is a protocol paper, some of the work has already started due to strict timing of funding, so we will aim to respond accordingly.

This cohort study will attempt to use multiple data streams to look at factors associated with child stunting in three countries. The strength of the study is an integrated protocol in three countries. I think there are some unacknowledged weakness and methodological issues that could be clarified however.

1. There are no details on the size of the pregnancy cohort, and no power calculations. It is unknown what are the primary outcomes of interest and if there is sufficient sample size to observe meaningful differences in the outcomes of interest (either pooled across all three sites or at the country level). Two of the country sites have relatively low rates of stunting, which also raises concern about the adequacy of the sample and number of observations to see what needs to be seen.

Thank you for this comment. This protocol paper is part of a collection of protocol papers that are currently been submitted to the BMJ Pediatrics journal supplement. All the papers make up the different workstreams Action Against Stunting Study. The information regarding the reason for choosing the particular sites, the sample size of the three pregnancy cohort, the power calculations and the primary and secondary outcomes will be included in the main protocol paper (Heffernan, et al).

2. The statistical analysis plan is vague - for example will the team be building models with a dependent variable of stunted/not stunted? or continuous anthropometric indicator? The authors also say they will develop DAGs as part of their plan, but these should already be developed and presented, else there is a risk that some important data element will be left out during data collection!

The main protocol paper (Heffernan, et al) will include a detailed section on the statistical analysis. We have therefore decided to remove the statistical analysis section in this paper and will refer to the main protocol paper.

We have added the following to line 19, page 9

3. If authors plan to conduct repeat measures assessments of anthropometric data, then attention needs to be given to the difficulty with especially height-for-age Z scores analyzed within subjects over time. Height for age difference is better. Jef Leroy and others have written extensively about this and it should be addressed.

The details of the statistical analysis will be included in the main protocol paper (Heffernan, et al)

4. Although the authors are planning a lot of diverse data, most of it remains within fairly traditional nutritional/anthropometric categories. If the goal is to contribute definitely to the conversation of stunting then the planned data streams are unlikely to significantly move the needle. What about microbiome? Intestinal mucosal dysfunction/ EED? Environmental toxins like aflatoxin? These are all active areas of investigation in the stunting field right now but are not included here. There also doesn't appear to be any plan to address WASH determinants, a major weakness.

Thank you for this comment. The Action Against Stunting Hub incorporates interdisciplinary workstreams with are conducting the work on the specific aspects you mention. These include the protocol paper on gut health (including intestinal mucosal dysfunction/EED) and the microbiome by Kadia et al., WASH by Haesler et al and epigenetics by Haggarty et al.

5. The major determinant of stunting is intergenerational/epigenetic effects. The plan to investigate this here is very limited, as far as I can tell only parental heights will be recorded. No DNA methylation or similar studies are proposed - this omission should be addressed.

As well as parental heights, saliva samples from the mother, father and infant will be taken at birth to analyse DNA methylation. Details of this are in the Epigenetics protocol paper by Haggarty et al.

6. From the perspective of intergenerational transmission of stunting, the timing for recruitment of the pregnancy cohort is also much too late. Opportunities to investigate preconceptional maternal determinants are missed, as is access to nutrients, adequate weight gain, and medical care during the critical early pregnancy period.

We fully agree with the reviewer that assessment of pre-conceptional maternal determinants is a strong study design. However, it was not feasible, for this cohort study undertaken in three countries and following children up to 24-months of age given the time frame and resources available. Also, recruitment in the first trimester would not have been appropriate given cultural sensitivities, in some

participating countries, of admitting pregnancy in the first trimester. The collection of data during the 2nd and 3rd trimester of pregnancy, however, will provide a wealth of relevant data related to exposures during pregnancy that will influence foetal and later infant growth, including a measure of weight gain in pregnancy, stress (hair cortisol reflects stress for the past 3 months) and maternal dietary intakes.

7. I have a major concern with the ultrasound protocol. I assume the primary goal here is to assess for fetal growth restriction. However, since the cohort is being recruited late in the second or third trimester, and no details are provided about pregnancy prior to that period, the value of this assessment is very limited. It is quite likely that many women will not know the date of the last menstrual period accurately. Without a first trimester dating ultrasound, and without accurate dates and scrupulous first trimester pregnancy care, the value of a later fetal ultrasound is very limited and will result in significant under- and over-estimation of fetal age/size. The mention of crown-rump length would also be irrelevant in a third trimester ultrasound.

We agree that first trimester scans are more reliable in establishing the gestational age and estimated fetal weight accurately but it also has its own limitations 1, 2. In our cohort, since we enrolled the participants in the second (Indonesia and Senegal) and the third trimester (India) of pregnancy, the study teams could only have US scans during the second and/or third trimesters as a part of the study protocol. However, data from the first and second trimester scans are available in around 80% and 90% participants, respectively. The first trimester scans have been used for gestational age estimation.

Studies have shown that routine ultrasound scan at around 32 weeks will help in detection of early Intrauterine growth restriction (IUGR) 3, 4. Therefore, within our limitations to perform the ultrasound scan at 2nd and third trimesters as a part of the study protocol, we think these ultrasound measurements will be useful to identify early IUGR by comparison with INTERGROWTH-21 charts. Moreover, the study will be able to evaluate the intrauterine growth velocities based on the data from the first and second trimester scan reports shared by the study participants and the scans done as a part of the study in around 80-90% participants. We think that this information, coupled with the infant growth data would be valuable because such data are scarce in low resource setting.

References:

Salomon LJ, Bernard JP, Duyme M, Doris B, Mas N, Ville Y. Feasibility and reproducibility of an image-scoring method for quality control of fetal biometry in the second trimester. *Ultrasound ObstetGynecol* 2006; 27: 34– 40.

Sladkevicius P, Saltvedt S, Almstrom H, Kublickas M, Grunewald C, Valentin L. Ultrasound dating at 12–14 weeks of gestation. A prospective cross-validation of established dating formulae in in-vitro fertilized pregnancies. *Ultrasound ObstetGynecol* 2005; 26: 504– 11.

Gordijn SJ, Beune IM, Thilaganathan B, Papageorgiou A, Baschat AA, Baker PN, Silver RM, Wynia K, Ganzevoort W. Consensus definition of fetal growth restriction: a Delphi procedure. *Ultrasound Obstet Gynecol*. 2016; 3(48):333–339

Sovio U, White IR, Dacey A, Pasupathy D, Smith GCS. Screening for fetal growth restriction with universal third trimester ultrasonography in nulliparous women in the Pregnancy Outcome Prediction (POP) study: a prospective cohort study. *Lancet*. 2015 Nov 21;386(10008):2089-2097. doi: 10.1016/S0140-6736(15)00131-2. Epub 2015 Sep 7. Erratum in: *Lancet*. 2015 Nov 21;386(10008):2058. PMID: 26360240; PMCID: PMC4655320.

8. For dietary assessments, it is unclear if efforts will be made to tabulate infant diet indicators in a way that would permit comparison to published WHO IYCF indicators.

The dietary data collected from the infants will allow estimation of the following WHO IYCF indicators:

These were not originally tabulated in Table 2, because they are secondary outcome indicators given the collection of quantitative 24-hour recall data. However, in response to the reviewer's comment, in Table 2, we have indicated the WHO IYCF indicators of breastfeeding (ever breastfed, early initiation of breastfeeding, exclusively breastfed for the first two days after birth, exclusive breastfeeding under six months, mixed milk feeding under six months and continued breastfeeding 12–23 months) and complementary feeding (introduction of solid, semi-solid or soft foods 6–8 months, minimum dietary diversity 6–23 months, minimum meal frequency 6–23 months, minimum milk feeding frequency for non-breastfed children 6–23 months, minimum acceptable diet 6–23 months, egg and/or flesh food consumption 6–23 months, sweet beverage consumption 6–23 months, unhealthy food consumption 6–23 months and zero vegetable or fruit consumption 6–23 months) and bottle feeding 0–23 months will be generated from the data collected (WHO, 2021).

Reference: WHO/UNICEF (2021). Indicators for assessing infant and young child feeding practices Definitions and measurement methods.

VERSION 2 – REVIEW

REVIEWER	Reviewer Name: Dr. Peter Flom Institution and Country: Peter Flom Consulting, United States
REVIEW RETURNED	03-Jan-2023

GENERAL COMMENTS	The authors have addressed my concerns and I now recommend publication.
---

REVIEWER	Reviewer Name: Dr. Peter Rohloff Institution and Country: Guatemala
REVIEW RETURNED	03-Jan-2023

GENERAL COMMENTS	Although most of the reviewer comments are well addressed in the author response letter, these are mostly not reflected in the paper. I would suggest the authors put a little more effort into making this a stand-alone paper to some degree, with at least small crossreferences to all the other papers in the collection. Readers may download and read this paper in isolation. The authors also mention constraints such as study timeline (study already started) and difference in the sites to the reviewers but not in the manuscript. Timing of enrollment and other limitations should be mentioned in a limitations section. Some of the reasoning provided to authors about the utility of the ultrasound data should be provided in the manuscript.
---

VERSION 2 – AUTHOR RESPONSE

Dear Editor of BMJ Pediatrics Open

Thank for returning our manuscript to revise our comments for reviewer 2. In order to respond to these comments accordingly, we request that we be allowed an addition of word count and references as reviewer 1 has accepted the manuscript as is, so it would be inappropriate remove any of the sections.

Reviewer 2's revised comments

Although most of the reviewer comments are well addressed in the author response letter, these are mostly not reflected in the paper. I would suggest the authors put a little more effort into making this a stand-alone paper to some degree, with at least small cross references to all the other papers in the collection.

Readers may download and read this paper in isolation. The authors also mention constraints such as study timeline (study already started) and difference in the sites to the reviewers but not in the manuscript. Timing of enrolment and other limitations should be mentioned in a limitations section. Some of the reasoning provided to authors about the utility of the ultrasound data should be provided in the manuscript.

Response to reviewer 2's revised comments

Thank you for your comments and useful suggestion that the paper should be able to be read in isolation. We agree that this will improve the paper, however, in order to do this, we have requested that the editor allows an increased word count.

1) To address the original comment 1 (at the end of responses), we have added the size of each cohort in line 3, page 5. "Prospective pregnancy cohort studies will be conducted in Hyderabad, India (n=717), East Lombok, Indonesia (n=702) and Kaffrine, Senegal (n=763)."

We have also added what the sample size and power calculation was based on line 4, page 5. "The sample size and power were calculated to test 100 epigenetic signatures (Epigenetic Studies in Children at Risk of Stunting and their Parents in India, Indonesia, and Senegal: a UKRI GCRF Action Against Stunting Hub protocol paper BMJ Paeds Open Ramsteijn et al, in press)."

2) To address the original comments 2 and 3 (at the end of responses), we have included the following in line 2, page 10: "The details of the statistical analysis of the Action Against Stunting Hub Study is explained in the main protocol paper (Developing a whole child approach for understanding stunting in early childhood: Introducing the UKRI GCRF Action Against Stunting Hub.Protocol Paper. BMJ Paeds Open Heffernan et al, in press). Data will firstly be analysed at country level and secondly pooled together where appropriate. The primary outcomes are: 1) growth velocity between time points from birth to 18 months and 2) height for age z scores at timepoints 3, 6, 9, 12 and 18 months. Based on the literature, we have decided a priori to include the following parental covariates in our models; age, cast (India only), village/health post, socioeconomic status, marital status, current work status and education status. Multivariable statistical models will be developed using Directed Acyclic Graphs and related approaches with pre-analysis to understand confounders and mediators on the pathway between exposures and outcomes. We will build our models and adjust for confounders using a forward stepwise approach with Akaike Information Criterion to identify the model of best fit (43). All analysis will be conducted using STATA version 18."

We have also included if variable or continuous, categorical or binary in Table 2, page 14.

3) To address the original comments 4 and 5 (at the end of responses), we have added information about the other data being collected by different interdisciplinary workstreams within AASH in line 24, page 9. "Data collected by other workstreams within the AASH study

The Action Against Stunting Hub incorporates interdisciplinary workstreams and will be collecting data on the following; detailed household questionnaires, epigenetics (mother, father and infant saliva samples), gut health and the microbiome (mother and infant stool samples), Water, Sanitation and Hygiene in the home environment (household observations), food systems (market and household observations and surveys) and education and cognition (household and pre-school observations) (Developing a whole child approach for understanding stunting in early childhood: Introducing the UKRI GCRF Action Against Stunting Hub.Protocol Paper. BMJ Paeds Open Heffernan et al, in press)"

4) To address the original comment 6 (at the end of responses), we have added the following to the limitations section to explain the reason for the timing of recruitment for the study in line 1, page 11.

“Firstly, it was not feasible, for this cohort study to include the preconception period given the time frame and resources available. Also, recruitment in the first trimester would not have been appropriate given cultural sensitivities, in some participating countries, of admitting pregnancy in the first trimester. The collection of data during the 2nd and 3rd trimester of pregnancy, however, will provide a wealth of relevant data related to exposures during pregnancy that will influence fetal and later infant growth, including a measure of weight gain in pregnancy, stress (hair cortisol reflects stress for the past 3 months) and maternal dietary intakes”

5) To address the original comment 7 (at end of responses), we have added the following to the ultrasound data section in line 11, page11

“Thirdly, in our cohort, since we enrolled the participants in the second (Indonesia and Senegal) and the third trimester (India) of pregnancy, the study teams could only carry out ultrasound scans during at these time-points rather than the first trimester. However, data from the first and second trimester scans are available in around 80% and 90% participants in India, respectively and will be utilized for gestational age estimation if available. If not, the date of last menstruation will be used. Studies have shown that routine ultrasound scans at around 32 weeks will help in detection of early Intrauterine growth restriction (IUGR) (44). Therefore, within our limitations to perform the ultrasound scan at 2nd and third trimesters as a part of the study protocol, we think these ultrasound measurements will be useful to identify early IUGR by comparison with INTERGROWTH-21 charts. This information, coupled with the infant growth data will be valuable because such data are scarce in low resource setting (45).”

Original comments from Reviewer 2:

1. There are no details on the size of the pregnancy cohort, and no power calculations. It is unknown what are the primary outcomes of interest and if there is sufficient sample size to observe meaningful differences in the outcomes of interest (either pooled across all three sites or at the country level). Two of the country sites have relatively low rates of stunting, which also raises concern about the adequacy of the sample and number of observations to see what needs to be seen.

2. The statistical analysis plan is vague - for example will the team be building models with a dependent variable of stunted/not stunted? or continuous anthropometric indicator? The authors also say they will develop DAGs as part of their plan, but these should already be developed and presented, else there is a risk that some important data element will be left out during data collection!

3. If authors plan to conduct repeat measures assessments of anthropometric data, then attention needs to be given to the difficulty with especially height-for-age Z scores analyzed within subjects over time. Height for age difference is better. Jef Leroy and others have written extensively about this and it should be addressed.

4. Although the authors are planning a lot of diverse data, most of it remains within fairly traditional nutritional/anthropometric categories. If the goal is to contribute definitely to the conversation of stunting then the planned data streams are unlikely to significantly move the needle. What about microbiome? Intestinal mucosal dysfunction/ EED? Environmental toxins like aflatoxin? These are all active areas of investigation in the stunting field right now but are not included here. There also doesn't appear to be any plan to address WASH determinants, a major weakness.

5. The major determinant of stunting is intergenerational/epigenetic effects. The plan to investigate this here is very limited, as far as I can tell only parental heights will be recorded. No DNA methylation or similar studies are proposed - this omission should be addressed.

6. From the perspective of intergenerational transmission of stunting, the timing for recruitment of the pregnancy cohort is also much too late. Opportunities to investigate preconceptional maternal determinants are missed, as is access to nutrients, adequate weight gain, and medical care during the critical early pregnancy period.

7. I have a major concern with the ultrasound protocol. I assume the primary goal here is to assess for fetal growth restriction. However, since the cohort is being recruited late in the second or third trimester, and no details are provided about pregnancy prior to that period, the value of this assessment is very limited. It is quite likely that many women will not know the date of the last menstrual period accurately. Without a first trimester dating ultrasound, and without accurate dates and scrupulous first trimester pregnancy care, the value of a later fetal ultrasound is very limited and will result in significant under- and over-estimation of fetal age/size. The mention of crown-rump length would also be irrelevant in a third trimester ultrasound.

VERSION 3 – REVIEW

REVIEWER	Reviewer Name: Dr. Peter Rohloff Institution and Country: Guatemala
REVIEW RETURNED	22-Mar-2023

GENERAL COMMENTS	With the additions the paper is more complete and stands alone.
---

REVIEWER	Reviewer Name: Dr. Peter Flom Institution and Country: Peter Flom Consulting, United States
REVIEW RETURNED	22-Mar-2023

GENERAL COMMENTS	The authors have addressed my concerns and I now recommend publication.
---

VERSION 3 – AUTHOR RESPONSE

N/A